# Perceptions of Harmfulness of Heated Tobacco Products Compared to Combustible Cigarettes among Adult Smokers in Japan: Findings from the 2018 ITC Japan Survey

**DOI:** 10.3390/ijerph17072394

**Published:** 2020-04-01

**Authors:** Shannon Gravely, Geoffrey T. Fong, Edward Sutanto, Ruth Loewen, Janine Ouimet, Steve S. Xu, Anne C. K. Quah, Mary E. Thompson, Christian Boudreau, Grace Li, Maciej L. Goniewicz, Itsuro Yoshimi, Yumiko Mochizuki, Tara Elton-Marshall, James F. Thrasher, Takahiro Tabuchi

**Affiliations:** 1Department of Psychology, University of Waterloo, Waterloo, ON N2L 3G1, Canadarloewen@uwaterloo.ca (R.L.); j2ouimet@uwaterloo.ca (J.O.); s4xu@uwaterloo.ca (S.S.X.); ackquah@uwaterloo.ca (A.C.K.Q.); y47li@uwaterloo.ca (G.L.); 2Ontario Institute for Cancer Research, Toronto, ON M5G 0A3, Canada; 3Division of Cancer Prevention and Population Sciences, Department of Health Behaviors, Roswell Park Comprehensive Cancer Center, Buffalo, NY 14263, USA; edward.sutanto@roswellpark.org (E.S.); maciej.goniewicz@roswellpark.org (M.L.G.); 4Department of Statistics and Actuarial Science, University of Waterloo, Waterloo, ON N2L 3G1, Canada; methompson@uwaterloo.ca (M.E.T.); cboudreau@uwaterloo.ca (C.B.); 5Division of Tobacco Policy Research, National Cancer Center Japan, 5-1-1 Tsukiji, Chuo-ku, Tokyo 104-0045, Japan; iyoshimi@ncc.go.jp; 6Japan Cancer Society, 13th Floor, Yurakucho Center Bldg. 2-5-1, Yurakucho, Chiyoda-ku, Tokyo 100-0006, Japan; mochizuki@jcancer.jp; 7Institute for Mental Health Policy Research, Centre for Addiction and Mental Health, London, ON N6G 4X8, Canada; Tara.EltonMarshall@camh.ca; 8Dalla Lana School of Public Health, University of Toronto, Toronto, ON M5T 3M7, Canada; 9Department of Epidemiology and Biostatistics, Schulich School of Medicine and Dentistry, Western University, London, ON N6A 5C1, Canada; 10Ontario Tobacco Research Unit, Toronto, ON M5S 2S1, Canada; 11Department of Health Promotion, Education and Behavior, Arnold School of Public Health, University of South Carolina, Columbia, SC 29208, USA; THRASHER@mailbox.sc.edu; 12Tobacco Research Department, Center for Population Health Research, National Institute of Public Health, Cuernavaca, Morelos 62100, Mexico; 13Cancer Control Center, Osaka International Cancer Institute, Chome-1-69 Otemae, Chuo Ward, Osaka 541-8567, Japan; tabuchitak@gmail.com

**Keywords:** heated tobacco products, heat-not-burn, modified risk tobacco products, combustible cigarettes, perceptions of harm, risk

## Abstract

In Japan, the tobacco industry promotes heated tobacco products (HTPs) as a reduced-risk tobacco product. This study examines: (1) smokers’ harm perceptions of HTPs relative to combustible cigarettes; (2) differences in relative harm perceptions between exclusive smokers and smokers who use HTPs (concurrent users) and between concurrent users based on frequency of product use; and (3) if smokers who were exposed to HTP advertising hold beliefs that are consistent with marketing messages of lower harmfulness. This cross-sectional study included 2614 adult exclusive cigarette smokers and 986 concurrent users who reported their perceptions of harmfulness of HTPs compared to cigarettes, as well as their exposure to HTP advertising in the last six months. Among all smokers, 47.5% perceive that HTPs are less harmful than cigarettes, 24.6% perceive HTPs to be equally as harmful, 1.8% perceive HTPs as more harmful, and 26.1% did not know. Concurrent users are more likely than exclusive smokers to believe that HTPs are less harmful (62.1% versus 43.8%, *p* < 0.0001) and less likely to report that they did not know (14.3% versus 29.4%, *p* < 0.0001). Frequent HTP users are more likely than infrequent users to believe that HTPs are less harmful (71.7% versus 57.1%, *p* ≤ 0.001). Believing that HTPs are less harmful than cigarettes was associated with noticing HTP advertising on TV (*p* = 0.0005), in newspapers/magazines (*p* = 0.0001), on posters/billboards (*p* < 0.0001), in stores where tobacco (*p* < 0.0001) or where HTPs (*p* < 0.0001) are sold, on social media (*p* < 0.0001), or in bars/pubs (*p* = 0.04). HTP users were significantly more likely than non-HTP users to believe that HTPs are less harmful than cigarettes, with this belief being more prominent among frequent users. Smokers who have been exposed to HTP advertising were more likely to perceive HTPs as less harmful than cigarettes.

## 1. Introduction

Smoked tobacco is the most dangerous and common form of nicotine consumption [1,2]. The smoke from cigarettes includes over 4000 chemicals and at least 70 known carcinogens [3]. Tobacco harm reduction is a public health strategy that aims to lower exposure to the toxicants produced by tobacco smoke, by encouraging smokers to completely substitute cigarettes with a less harmful product that can deliver nicotine. Alternative nicotine products, such as e-cigarettes and snus, can effectively deliver nicotine without combustion, and are considered to be less harmful than cigarettes [4,5,6,7]. Notably, the complete substitution of reduced-risk alternatives for cigarettes may offer a substantial reduction in smoking prevalence and mortality if widely adopted [8,9].

Heated tobacco products (HTPs) are an example of an alternative tobacco product that has been developed by the tobacco industry and may be potentially less harmful than cigarettes [6,7,10,11,12,13]. HTPs heat tobacco in loose leaf form or contained in tobacco sticks, plugs, or capsules using a battery-powered heating system. Tobacco is heated (not combusted like traditional cigarettes) to generate an inhalable nicotine-containing aerosol [12,14].

HTPs are not a new concept. They were first introduced in the 1980s, but unsuccessfully [12]. A newer generation of HTPs (e.g., IQOS from Philip Morris International (PMI)) has been introduced in many countries around the world. HTP market growth has generally been slow in most countries, with Japan being an exception [15]. HTPs were introduced into Japan’s open market in 2016, initially with IQOS, which was followed by Ploom TECH from Japan Tobacco and glo from British American Tobacco in 2017. According to a market report by PMI, IQOS had captured 17.4% of Japan’s tobacco share in 2019 [16], and although Japan’s HTP market has expanded with other HTPs, IQOS still controls about 70% of the HTP market in Japan [16]. The growing popularity of HTPs in Japan (accounting for 24.3% of Japan’s tobacco market in March 2019 [16]) may in part be due to the prohibition of nicotine e-cigarettes; thus, there is no competition between HTPs and e-cigarettes.

The tobacco industry has been actively marketing HTPs in Japan as a reduced-risk tobacco product, due to its purported non-combustible nature [11]. However, there is currently no scientific consensus that HTPs are safer than cigarettes [12,13,14], or if they will be a successful smoking cessation aid [14]. Some comprehensive reviews of the literature have suggested that HTPs may be less harmful than cigarettes because they are likely to expose users and bystanders to lower levels of particulate matter and harmful compounds [7,12,13,17], but the World Health Organization (WHO) strongly debates that they should be used as an alternative to cigarettes based on the reduced-risk assertion made by the tobacco industry [18]. Despite the current scientific uncertainty about HTPs’ absolute or relative harmfulness compared to cigarettes (mainly owing to large variability of reduced toxicant profiles between studies) [7], Japanese law does not prevent them from being marketed as a reduced-risk product.

It is currently unclear how smokers in Japan perceive the harmfulness of HTPs relative to cigarettes and how exposure to marketing may be associated with harmfulness perceptions. Thus, the aims of this study were to examine: (1) smokers’ harm perceptions of HTPs relative to cigarettes; (2) differences in relative harm perceptions between: (i) exclusive smokers and smokers who use HTPs (concurrent users); (ii) concurrent users based on HTP use frequency (frequent versus infrequent HTP users); and (iii) concurrent users based on HTP and smoking frequency; and (3) whether smokers’ beliefs about lower harmfulness are associated with exposure to HTP advertising.

## 2. Methods

### 2.1. Sample, Study Design, and Procedure

The International Tobacco Control (ITC) Wave 1 Japan Survey (conducted February–March 2018) was a web-administered survey of behaviors and attitudes related to tobacco and nicotine use among a nationally representative sample of 4615 adult (aged 20+) Japanese exclusive cigarette smokers (smoke at least monthly), exclusive HTP users (use HTPs at least weekly), concurrent HTP users and cigarette smokers, and non-users (do not smoke or use HTPs). The survey was conducted with Rakuten Insight’s proprietary online panel in Japan, with quotas for region of residence, gender, and age, to ensure that the final sample was proportional to stratum sizes from Japan census data. Recruitment for the panel was conducted on a daily basis, tapping into users of Rakuten services (e.g., e-commerce, credit cards, insurance, mobile services, etc.), as well as other online resources such as affiliates, email and banner recruits in order to maintain a panel as consistent as possible with the general population. Panelists received email invitations and also had the option of logging into their proprietary panel site to access the survey.

The overall survey response rate was 45.1% (with a 96.3% cooperation rate). The survey protocols and all materials, including the survey questionnaires, were cleared for ethics by Office of Research Ethics, University of Waterloo, Canada (ORE#22508/31428). All participants provided consent to participate. Further methodological details can be found in the Wave 1 ITC Japan technical report (Japan Wave 1 Technical Report).

For the present cross-sectional analyses, respondents were included if, at the time of the survey they: (1) had ever heard of HTPs; (2) were current cigarette smokers (smoked at least monthly); and (3) had complete outcome data for the questions about harmfulness of HTPs compared to cigarettes, and exposure to HTP marketing. Of the overall sample of 4615 Wave 1 respondents, 3838 met the inclusion criteria of being a current (at-least monthly) smoker. Of these, 3600 respondents had heard of HTPs and completed the relevant outcome questions; 2614 were exclusive cigarette smokers; and 986 were concurrent users (see the Study Flow Diagram: Appendix A).

### 2.2. Measures

The survey, with original response options, can be found at the ITC Project website: ITC JP 1 Survey. The following variables were used in the current study:

#### 2.2.1. Sociodemographic Variables

Sociodemographic measures were: gender (female versus male), age group (20–29, 30–39, 40–59, ≥60 years), income [low: 4,000,000 yen or less; medium: more than 4,000,000 yen to 6,000,000 yen; high: more than 6,000,000 yen; or not reported], and education [low: junior high school, high school; medium: vocational school, junior college/ technical college; high: undergraduate or higher; or not reported].

#### 2.2.2. Smoking Variables

Respondents reported their cigarette smoking frequency at the time of the survey and were categorized as a daily or non-daily (weekly or monthly) smoker.

#### 2.2.3. HTP Use Variables

Respondents reported whether they had ever heard of HTPs: ‘yes’ or ‘no’; those who reported that they did not know were classified as ‘no’. If respondents had heard of HTPs they were asked: “Have you ever used one of these heat-not-burn products, even one time?” (‘yes’ or ‘no’). If respondents reported ever using an HTP, they were asked about frequency of use: “How often do you CURRENTLY use heat-not-burn products? These include products such as IQOS, Ploom TECH, and glo”. Response option were: daily, weekly, monthly, less than monthly, or not at all. Current HTP use was classified as: ‘any current HTP use’ (daily, weekly, monthly, or less than monthly) or ‘not currently using an HTP’ (tried it/used it in the past, or never tried one).

#### 2.2.4. Smoker and/or HTP User Groups

Smokers were divided into two main user groups:Exclusive Smoker: currently smokes cigarettes at least monthly AND does not currently use an HTP.Concurrent User: currently smokes cigarettes at least monthly AND currently uses an HTP (any current HTP use).

#### 2.2.5. Concurrent User Groups

Concurrent users were further categorized using two methods, based on frequency of smoking and HTP use:HTP frequency only: ‘Frequent HTP user’ (at least weekly use) versus ‘Infrequent HTP user’ (less than weekly use).HTP and smoking frequency, using the classification system derived in Borland et al. (because daily nicotine users differ considerably from non-daily nicotine users) [19]: (i) ‘Predominant smoker’: Daily smoking and less-than-daily HTP use; (ii) ‘Concurrent-daily user’: Daily smoking and daily HTP use; (iii) ‘Concurrent non-daily user’: Less-than-daily smoking and less-than-daily HTP use; or (iv) ‘Predominant HTP user’: Less than daily smoking and daily HTP use.

#### 2.2.6. Perceived Harmfulness of HTPs Relative to Cigarettes

Respondents who were aware of HTPs were asked the following question: “Compared to smoking cigarettes, how harmful do you think using a heat-not-burn tobacco product is?”. Responses options included: ‘Much less harmful than smoking cigarettes’, ‘Somewhat less harmful than smoking cigarettes’, ‘Equally harmful to smoking cigarettes’, ‘Somewhat more harmful than smoking cigarettes’, or ‘I do not know’. These responses were re-categorized as: ‘HTPs are less harmful than cigarettes’, ‘HTPs are equally harmful as cigarettes’, ‘HTPs are more harmful than cigarettes’, or ‘I do not know’.

#### 2.2.7. Exposure to HTP Ads Via Various Marketing Platforms

Respondents answered the question “In the last six months have you noticed heat-not-burn products being advertised in any of the following places?”: TV; radio; newspapers/magazines; posters/billboards; stores where tobacco products are sold; stores where HTPs are sold; social media; and bars/pubs. Responses were categorized as: ‘yes’ versus ‘no/ I do not know’.

### 2.3. Statistical Analyses

Unweighted data were used to describe the study sample. Chi-square tests were conducted overall and between the user groups (exclusive smokers versus concurrent users). All subsequent analyses were conducted using weighted data. Cross-sectional weights were computed for all respondents. A raking algorithm was used to calibrate the weights on smoking status, HTP use, geographic region, and demographic measures (e.g., gender, age, ethnicity, and education) [20].

In the first set of analyses, two multinomial regression models were conducted to compute weighted and adjusted estimates for perceived relative harmfulness of HTPs compared to cigarettes (HTPs are less harmful versus equally harmful versus more harmful versus I do not know). Model 1 (objective 1) included all respondents and controlled for: gender, age, income, education, smoking frequency (daily versus non-daily) and frequency of HTP use (daily versus non-daily versus not using one). In Model 2 (objective 2 (i)), perceptions of harmfulness were examined by user group status (exclusive smokers versus concurrent users). The model controlled for: gender, age, income, education, and smoking frequency (daily versus non-daily). A sub-analysis was conducted using adjusted weighted logistic regression models to estimate the odds of believing that HTPs are less harmful than cigarettes (compared to equally harmful/more harmful/I do not know) for all respondents, as well as broken down by user group: exclusive smokers and concurrent users.

The second set of analyses was restricted to the concurrent user group, those who both smoke and use HTPs. The first multinomial regression (objective 2 (ii)) compared perceptions of relative harmfulness between frequent HTP users (at least weekly use) and infrequent HTP users (less than weekly use). The outcome was re-categorized as: less harmful versus more/equally harmful versus I do not know, in order to account for low proportion of respondents who reported that HTPs were more harmful than cigarettes (2.4%). The model controlled for gender, age, income, education, and smoking frequency (daily versus non-daily). The second multinomial regression model (objective 2 (iii)) examined difference between concurrent users based on smoking and HTP use frequency and compared: predominant smokers versus concurrent-daily users versus concurrent non-daily users. Due to the small sample size of concurrent non-daily users (*n* = 58), we used a dichotomized outcome (HTPs are less harmful versus HTPs are equally/more harmful/I do not know). Predominant HTP users were excluded due to the small sample size (*n* = 4). The model controlled for gender, age, income, and education.

The final set of analyses (objective 3) included all respondents and compared perceptions of relative harm of HTPs (HTPs are less harmful than cigarettes versus equally/more/I do not know) for those who had or had not noticed HTP advertising via various marketing platforms (yes versus no/I do not know). The weighted logistic regression models controlled for gender, age, income, education, and user group (exclusive smokers versus concurrent users).

Statistical significance and confidence intervals were tested at the 95% confidence level. Analyses were conducted using SAS Version 9.4 (SAS Institute Inc. 2013, Cary, North Carolina, USA).

## 3. Results

Table 1 presents the (unweighted) respondents’ characteristics (overall and by user group status). Overall, 76.4% of the total sample were male, 94.4% were daily cigarette smokers, and 11.1% were current daily HTP users. The majority of the sample was aged 40+ (67.2%), with concurrent users being significantly younger than exclusive smokers (*p* < 0.0001). Among the exclusive smokers (*n* = 2614), 94.7% smoked daily; among concurrent users (*n* = 986), 93.7% smoked daily, and 40.6% used HTPs daily.

### 3.1. Smokers’ Perceptions of Harmfulness of HTPs Relative to Cigarettes

Of the 3600 smokers, 47.5% perceived that HTPs were less harmful than cigarettes, 24.6% perceived HTPs to be equally as harmful than cigarettes, 1.8% perceived HTPs as more harmful, and 26.1% did not know (Figure 1 and Appendix A).

#### Differences in Relative Harm Perceptions between Exclusive Smokers and Concurrent Users

When user groups were compared, concurrent users were significantly more likely than exclusive smokers to believe that HTPs are less harmful (62.1% versus 43.8%, *p* < 0.0001) and significantly less likely than exclusive smokers to report that they did not know (14.3% versus 29.4%, *p* < 0.0001) (Figure 1 and Appendix A).

### 3.2. Odds of Believing That HTPs Are Less Harmful than Cigarettes: Exclusive Smokers versus Concurrent Users

Concurrent users were twice as likely as exclusive smokers to believe that HTPs are less harmful, compared to being equally harmful, more harmful, or uncertain (odds ratio (OR) = 2.1, 95% confidence interval (CI): 1.7–2.5), which is likely being driven by the more frequent HTP users, who are more likely to perceive that HTPs are less harmful than infrequent HTP users (OR = 1.9, 95% CI: 1.4–2.6). Concurrent daily users were twice as likely as predominant smokers (OR = 0.5, 95% CI: 0.4–0.7) and as concurrent non-daily users (OR = 0.5, 95% CI: 0.2–0.9) to believe that HTPs are less harmful (Table 2).

#### 3.2.1. Differences in Relative Harm Perceptions between Concurrent Users

Differences in relative harm perceptions by HTP use frequency: When concurrent users were categorized by frequency of HTP use, frequent users were significantly more likely than infrequent users to believe that HTPs are less harmful than cigarettes (71.7% versus 57.1%, *p* < 0.001). More of the infrequent HTP users reported that they did not know, compared to frequent users (13.6% versus 9.0%, *p* < 0.01) (Figure 2 and Appendix A).

Differences in relative harm perceptions by frequency of HTP use and smoking: When concurrent users were compared by the frequency of HTP use and smoking, the analysis showed there was a main effect between these groups (*p* = 0.0005). Concurrent daily users were more likely to believe HTPs are less harmful than both predominant smokers (73.2% versus 58.8%, *p* < 0.001) and concurrent non-daily users (56.4%, *p* = 0.03) (Figure 3 and Appendix A).

#### 3.2.2. Exposure to Various Marketing Platforms that Advertise HTPs

Smokers who reported noticing (e.g., were exposed to) HTP marketing in the last six months, compared to those who did not notice, were more likely to believe that HTPs are less harmful, for nearly all venues where marketing was observed: TV (OR = 1.4, 95% CI: 1.1–1.6, *p* < 0.001), newspapers or magazines (OR = 1.4, 95% CI: 1.2–1.7, *p* = 0.0001), posters or billboards (OR = 1.4, 95% CI: 1.2–1.7, *p* < 0.0001), stores where tobacco (OR = 1.7, 95% CI: 1.4–2.0, *p* < 0.0001) or HTPs (OR = 1.5, 95% CI: 1.3–1.8, *p* < 0.0001) are sold, social media (OR = 1.6, 95% CI: 1.3–1.9, *p* < 0.0001), and bars/pubs (OR = 1.3, 95% CI: 1.0–1.7, *p* = 0.045). There were no differences between those who noticed HTP advertising on the radio and those who did not (*p* = 0.35) (Table 3).

## 4. Discussion

The current study examined perceptions of harmfulness of HTPs compared to cigarettes among a sample of adult Japanese exclusive smokers and concurrent users of cigarettes and HTPs, finding that about half of smokers perceived HTPs as being less harmful than cigarettes. A quarter believed that they are equally as harmful and a quarter did not know. Very few smokers believed that HTPs are more harmful than cigarettes. Concurrent users were twice as likely than exclusive smokers to believe that HTPs are less harmful and were also less likely to report that they did not know. Among concurrent users only, frequent HTP users were twice as likely than infrequent users to believe that HTPs are less harmful, and concurrent daily users were twice as likely as predominant smokers and concurrent non-daily users to believe that HTPs are less harmful than cigarettes. Smokers exposed to HTP advertising via various marketing platforms, including TV, billboards, social media, newspapers, magazines, bars/pubs, and stores where tobacco and HTPs are sold, were more likely to perceive HTPs as less harmful than cigarettes.

Consumer perceptions of HTP harmfulness relative to cigarettes may be a key predictor of initiation and continued use of HTPs. For example, smokers are more likely to switch to or initiate use of other alternative nicotine products (e.g., e-cigarettes) if they believe them to be less harmful than cigarettes [21,22,23]; however, few studies have addressed this for HTPs. A recent qualitative study of current and former IQOS users in the United Kingdom found that one of the main reasons for initiating HTP use and continued use was because it is ‘less harmful’, ‘less hazardous’, and ‘less damaging’ for their health than cigarettes [24]. A cross-sectional online survey of HTP users from various countries, mainly from Switzerland, reported 89% of respondents were using HTPs because they are less toxic than cigarettes [25]. Two other research studies in Korea [26] and Germany [27] also reported HTP use was positively associated with believing that they were less harmful than cigarettes. Our results reflect these findings, such that those who reported using HTPs in our study were more likely than non-HTP-users to believe they are less harmful than cigarettes, and this belief was greater among the more frequent HTP users.

Advertising and other promotional approaches are critical to increasing the appeal and use of tobacco products [28]. For example, tobacco image advertising is designed to increase product visibility and to communicate appealing attributes such as attractiveness, performance, identity, and quality [28]. HTPs are no exception, especially in Japan where there are no advertising bans and few product packaging laws.

HTPs are marketed as sophisticated, modern, high tech, and clean reduced-risk products in order to increase appeal, and to minimize health concerns among Japanese consumers (IQOS Ads and Products) [11,18,29]. The tobacco industry has used creative marketing strategies via a large number of advertising platforms (e.g., stores, TV, billboards, magazines, and social media) in Japan to attract customers and increase HTP sales [29]. Although our data cannot determine if exposure to various types of HTP advertising impacted HTP relative harm perceptions, or even HTP use, we did find that exposure to HTP advertising was associated with the belief that HTPs are less harmful than cigarettes, and HTP users were more likely to hold this belief compared to non-HTP users.

Our study did not examine whether HTP users are former or never-smokers who started using HTPs because they believe they are less harmful alternative to cigarettes, or if current smokers want to substitute their cigarettes with a potentially less harmful alternative; however, some research has suggested that HTPs may in fact be barrier to smoking cessation in Japan. For example, despite PMI’s claims that most IQOS users are former smokers [16], independent research has shown that the majority of Japanese HTP users do not quit smoking [29,30,31,32]. For example, Xu et al. found that, among the reasons for using HTPs cited by Japanese concurrent users, non-quit reasons—to cut down on cigarettes (64.6%) and to not have to give up cigarettes altogether (52.1%)— were mentioned as often as or more often than the desire to quit smoking (55.4%) [33]. Research has consistently shown that light smoking [34] or reducing cigarette consumption (even among heavy smokers who reduce their daily consumption by 50%) does not result in a reduction of smoking-attributable disease and death [35], therefore using HTPs to simply reduce cigarette consumption should be strongly discouraged.

Although this large survey of Japanese smokers and concurrent users was nationally representative, some limitations exist. First, because this study was cross-sectional, it cannot be used to demonstrate causality. For example, we cannot determine whether perceptions of harm preceded or followed HTP use. Similarly, it is also not possible to determine if smokers changed their beliefs about the relative harm of HTPs compared to cigarettes because they were exposed to HTP marketing, or if they were better able to recall marketing because it was in line with their own beliefs. Longitudinal cohort studies are needed to examine the impact of HTP marketing on harm perceptions, attitudes, and beliefs, and if smokers who perceive HTPs to be less harmful will be more likely to initiate use, and how this may impact subsequent smoking cessation. Secondly, this study only included current smokers, therefore the results from this study cannot be generalizable to former smokers or non-smokers. Third, these results were obtained from adults in Japan, and thus may not apply to other countries or younger age groups.

## 5. Conclusions

Overall, this study found that nearly half of all smokers believed HTPs to be less harmful than cigarettes. HTP users were significantly more likely to believe that HTPs are less harmful than cigarettes compared to non-HTP users, with this belief being more prominent among frequent users. Smokers who have been exposed to HTP marketing appear to hold beliefs that are consistent with these advertising messages in Japan [36]. Even though existing scientific data show less exposure to toxic substances from HTPs than from cigarettes [7,12,13,17], advertising HTPs as reduced-risk products should be prohibited until the scientific community comes to a consensus about the ultimate effect of HTP use on health. Moreover, if the use of HTPs continues to increase in Japan, and possibly in other countries, the provision of balanced information and smoking cessation advice to smokers who are interested in using HTPs in place of combustible cigarettes is imperative. Communication should be framed to differentiate between relative and absolute harms, while providing an evidence-based appraisal of the relative risk of HTPs in comparison to combustible cigarettes.

## Figures and Tables

**Figure 1 ijerph-17-02394-f001:**
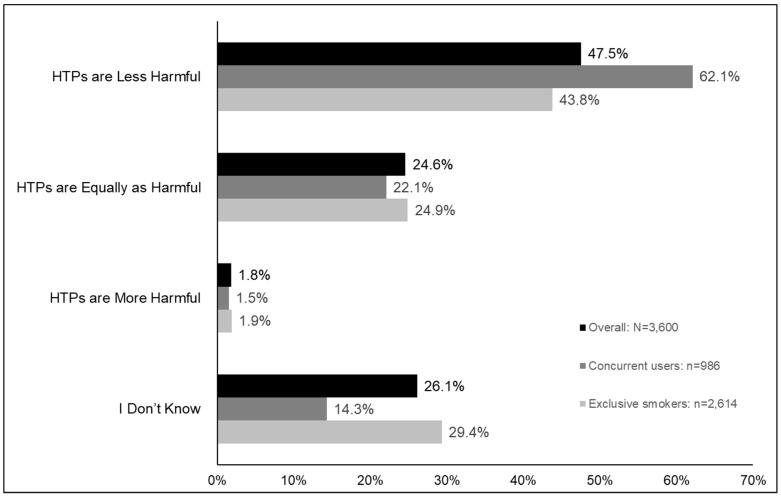
Relative perceptions of harmfulness of HTPs compared to cigarettes among smokers in Japan in 2018.

**Figure 2 ijerph-17-02394-f002:**
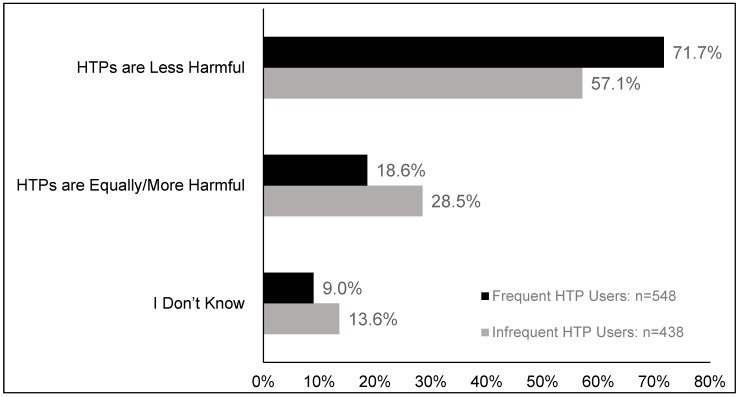
Relative perceptions of harmfulness of HTPs compared to cigarettes among concurrent users stratified by HTP use frequency (frequent HTP users versus infrequent HTP users). Data are weighted and adjusted. HTPs: Heated tobacco products. Frequent HTP user: at least weekly use; infrequent HTP user: less than weekly use.

**Figure 3 ijerph-17-02394-f003:**
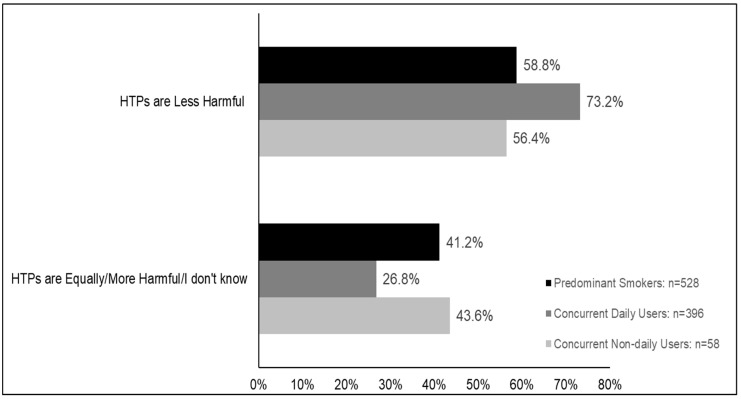
Relative perceptions of harmfulness of HTPs compared to cigarettes among concurrent users stratified by smoking and HTP use frequency (Predominant Smokers vs. Concurrent-daily Users vs. Concurrent Non-daily Users). Data are weighted and adjusted. HTPs: Heated tobacco products. Predominant Smokers: smoke cigarettes daily and use HTP less than daily; Concurrent-daily users: smoke cigarettes and use HTP daily; Concurrent non-daily users: smoke cigarettes and use HTP less than daily.

**Table 1 ijerph-17-02394-t001:** Unweighted sample characteristics of respondents at Wave 1 (2018).

Chacteristics		Exclusive Smokers *n* = 2614 (72.6)	Concurrent Users *n* = 986 (27.4)	Chi-Square (χ^2^) Test between User Groups	All Respondents *n* = 3600	Overall Chi-Square
		*n* (%)	*n* (%)		*n* (%)	
Gender	Male	2026 (77.5)	723 (73.3)	χ^2^ = 6.9, *p* = 0.009	2749 (76.4)	*p* < 0.0001
	Female	588 (22.5)	263 (26.7)		851 (23.6)	
Age group (years)	20-29	197 (7.5)	178 (18.1)	χ^2^ = 205.0, *p* < 0.0001	375 (10.4)	*p* < 0.0001
	30-39	494 (18.9)	310 (31.4)		804 (22.3)	
	40-59	1198 (45.8)	372 (37.7)		1570 (43.6)	
	≥60	725 (27.7)	126 (12.8)		851 (23.6)	
Education level	Low	993 (38.0)	318 (32.3)	χ^2^ = 52.4, *p* < 0.0001	1311 (36.4)	*p* < 0.0001
	Medium	435 (16.6)	168 (17.0)		603 (16.8)	
	High	1158 (44.3)	495 (50.2)		1653 (45.9)	
	Not reported	28 (1.1)	5 (0.5)		33 (0.9)	
Income	Low	727 (27.8)	189 (19.2)	χ^2^ = 14.5, *p* = 0.002	916 (25.4)	*p* < 0.0001
	Medium	571 (21.8)	247 (25.1)		818 (22.7)	
	High	956 (36.6)	457 (46.4)		1413 (39.3)	
	Not reported	360 (13.8)	93 (9.4)		453 (12.6)	
Smoking frequency	Daily	2476 (94.7)	924 (93.7)	χ^2^ = 1.4, *p* = 0.24	3400 (94.4)	*p* < 0.0001
	Non-daily	5.3 (5.3)	6.3 (6.3)		200 (5.6)	
HTP use frequency	Daily	—	400 (40.6)	N/A	400 (11.1)	*p* < 0.0001
	Weekly	—	148 (15.0)		148 (4.1)	
	Monthly	—	94 (9.5)		94 (2.6)	
	Less than monthly	—	344 (34.9)		344 (9.6)	
	Not at all	2614 (100.0)	—		2614 (72.6)	
Concurrent User Group	Predominant smoker	—	396 (40.2)	N/A	528 (14.7)	*p* < 0.0001
	Concurrent-daily user	—	528 (53.5)		396 (11.0)	
	Concurrent non-daily user	—	58 (5.9)		58 (1.6)	
	Predominant HTP user	—	4 (0.4)		4 (0.1)	

Predominant smokers: Smoke cigarettes daily and use HTPs less than daily; Concurrent-daily users: Smoke cigarettes and use an HTP daily; Concurrent non-daily users: does not smoke cigarettes or use an HTP daily; predominant HTP user: daily HTP use; non-daily smoker. HTP: Heated tobacco product. N/A: Not applicable.

**Table 2 ijerph-17-02394-t002:** Adjusted (Weighted) regression models estimating the odds of believing that HTPs are less harmful than cigarettes among smokers in Japan in 2018.

User Group All Users (*N =* 3600)	HTPs Are Less Harmful %	Odds Ratio	95% CI	
			Lower CI	Upper CI
Exclusive Smokers (*n* = 2614)	43.8%	Reference		
Concurrent Users (*n* = 986)	62.1%	2.1	1.7	2.5
Concurrent Users (*n* = 986)				
Frequent HTP Users (*n* = 548)	71.7%	1.9	1.4	2.6
Infrequent HTP Users (438)	57.1%	Reference		
Concurrent Users (*n* = 982)				
Predominant Smokers (*n* = 528)	58.8%	0.5	0.4	0.7
Concurrent Daily Users (*n* = 396)	73.2%	Reference		
Concurrent Non-daily Users (*n* = 58)	56.4%	0.5	0.2	0.9

Outcome: Less harmful versus equally/more harmful/do not know. HTPs: Heated tobacco products; CI: Confidence interval.

**Table 3 ijerph-17-02394-t003:** Adjusted (weighted) regression models estimating the odds of believing that HTPs are less harmful than cigarettes among smokers in Japan in 2018 based on exposure to various marketing platforms (*n* = 3600).

Advertising Location	Exposure	HTPs Are Less Harmful %	*p*-Value	Odds Ratio	95% CI	
					Lower CI	Upper CI
TV	Yes (*n* = 890)	52.1%	0.0005	1.4	1.1	1.6
	No (*n* = 2710)	44.3%		Reference		
Radio	Yes (*n* = 170)	50.5%	0.35	1.2	0.8	1.7
	No (*n* = 3430)	46.0%		Reference		
Newspapers or magazines	Yes (*n* = 1012)	52.3%	0.0001	1.4	1.2	1.7
	No (*n* = 2588)	43.9%		Reference		
Posters or billboards	Yes (*n* = 1366)	51.9%	<0.0001	1.4	1.2	1.7
	No (*n* = 2234)	42.9%		Reference		
Stores where tobacco is sold	Yes (*n* = 2394)	50.6%	<0.0001	1.7	1.4	2.0
	No (*n* = 1206)	38.3%		Reference		
Stores where HTPs are sold	Yes (*n* = 2164)	50.5%	<0.0001	1.5	1.3	1.8
	No (*n* = 1436)	40.5%		Reference		
Social media	Yes (*n* = 895)	54.5%	<0.0001	1.6	1.3	1.9
	No (*n* = 2705)	43.6%		Reference		
Bars or pubs	Yes (*n* = 367)	52.3%	0.045	1.3	1.0	1.7
	No (*n* = 3233)	45.6%		Reference		

Outcome: Less harmful versus equally/more harmful/do not know (Reference).

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
