# Peer review of "Perceptions of Harmfulness of Heated Tobacco Products Compared to Combustible Cigarettes among Adult Smokers in Japan: Findings from the 2018 ITC Japan Survey"

_ijerph, 2020, doi:10.3390/ijerph17072394_

Round 1

Reviewer 1 Report

Amongs other more specific objetives, this study examined smoker’s harm perceptions of heated tobacco products (HTP) relative to combustible tobacco cigarettes. It is a cross-sectional study based on 2,614 smokers and 986 concurrent users. This study is timely and relevant, specially as use of HTPs is increasing worldwide. Overall, most of the sample reported that HTPs are less harmful than cigarettes. Importantly, concurrent use of combustible cigarettes and HTPs were associated with lower harm perception. I'd like to thank the opportunity to review this study. Authors should only address minor issues which I stress below:

Methods:

1) Authors should specific what they mean by “low, medium or high” education (see last paragraph on page 19).

2) Results: Authors give some information on the number of participants that met the inclusion criteria. This should be moved to the methods section. The Supplementary Figure 1 should also be mentioned in the methods.

3) On Table 1, please add statistics and effect sizes. This information alongside p values should be appropriately provided in tables.

4) Please, remove Figure 1-2 and add instead supp tables in text (is more informative and accurate). On another note, please add information on effect sizes.

Author Response

Reviewer 1

Amongs other more specific objetives, this study examined smoker’s harm perceptions of heated tobacco products (HTP) relative to combustible tobacco cigarettes. It is a cross-sectional study based on 2,614 smokers and 986 concurrent users. This study is timely and relevant, specially as use of HTPs is increasing worldwide. Overall, most of the sample reported that HTPs are less harmful than cigarettes. Importantly, concurrent use of combustible cigarettes and HTPs were associated with lower harm perception. I'd like to thank the opportunity to review this study. Authors should only address minor issues which I stress below:

Authors’ Response: The authors thank the reviewer for their time and insightful feedback. Please find responses below to your peer-review of this paper.

Methods

  1. Authors should specific what they mean by “low, medium or high” education (see last paragraph on page 19).

Authors’ Response: A brief description of both education and income has been added to the Methods on page 4.

  1. Results: Authors give some information on the number of participants that met the inclusion criteria. This should be moved to the methods section. The Supplementary Figure 1 should also be mentioned in the methods.

Authors’ Response: The description of the eligible respondents (and reference to the Study Flow Diagram) was removed from the Results and added to the study description section on page 3.

  1. On Table 1, please add statistics and effect sizes. This information alongside p values should be appropriately provided in tables.

Authors’ Response: The authors have added the following analyses to Table 1: (i) a chi-square test between the User Groups (exclusive smokers vs concurrent users); and (2) a p-value for the overall chi-square. This has also been updated in the Statistical analyses section on page 5.

  1. Please, remove Figure 1-2 and add instead supp tables in text (is more informative and accurate). On another note, please add information on effect sizes.

Authors’ Response: The authors would like to request that the Reviewer reconsider the deletion of the tables. We believe that they help break up some of the extensive data shown herein. We have added effect sizes (ORs and 95% CIs) to the Supplemental tables.

Reviewer 2 Report

Abstract

  1. “…beliefs that are consistent with tobacco industry marketing claims that HTPs are less harmful than cigarettes” This wording implies that these claims are incorrect; I don’t think there is evidence indicating that HTPs are more harmful than combustible cigarettes but there is evidence they are less harmful, so this belief is not just consistent with tobacco industry claims but likely consistent with reality. Please reword
  2. “Smokers who have been exposed to tobacco industry marketing about HTPs appear to hold beliefs that are consistent with tobacco industry advertising messages.” It is not possible to say from the data here if these smokers changed their beliefs because they were exposed to marketing or if e.g. they were better able to recall marketing because it was in line with their beliefs. As this is a cross-sectional survey, it is not possible to make such a directional statement. This also applies to the discussion and conclusion.

Introduction

  1. ‘exploded onto the market’ strikes me as an unfortunate choice of words.
  2. It may be worth mentioning that e-cigarettes are not legal in Japan as this may be one factor in their success or in companies targeting the country specifically

Methods

  1. Please mention the minimum age for inclusion in the sample section (20 as suggested by the sociodemographics?)
  2. For the HTP user groups, I would suggest just using the response options as labels (e.g. at least weekly versus less than weekly).
  3. Can you link the analyses more clearly to the aims please, i.e. which analysis was run to address each aim. Similarly, the results sections could be structured more clearly along the aims, e.g. using short versions of the aims as headings

Results

  1. In the supplemental figure, the arrows for exclusions each sit one box too low.
  2. Spotted a typo in table 1 for 'age group 30-39'
  3. In Figure 1, it would be interesting to see the full range of response options from much less harmful to more harmful

Discussion

  1. Having pictures in the discussion of a scientific article is unusual and these do not seem to be crucial to the manuscript; I would strongly recommend removing Figures 4 and 5.
  2. The text on marketing (p12 to 13) is quite long and detailed, I am not convinced all of it is relevant to the present manuscript.
  3. What is the evidence that these products are not reduced risk products compared with cigarettes? The strong statements in the conclusion would need evidence to underpin them or need rewording.

Author Response

Reviewer 2

Authors’ Response: The authors thank the reviewer for their time and insightful feedback. Please find responses below to your peer-review of this paper.

Abstract

  1. “…beliefs that are consistent with tobacco industry marketing claims that HTPs are less harmful than cigarettes” This wording implies that these claims are incorrect; I don’t think there is evidence indicating that HTPs are more harmful than combustible cigarettes but there is evidence they are less harmful, so this belief is not just consistent with tobacco industry claims but likely consistent with reality. Please reword

Authors’ Response: The authors completely agree that there is some scientific evidence that HTPs are less harmful than cigarettes (although we do not know the extent of relative harmfulness yet). Most countries however do not allow the tobacco industry to market HTPs in this way. Notably, in Japan, most of the industry’s messaging, directly promotes HTPs as significantly less harmful to consumers. In the abstract and the main manuscript, we do not say that the tobacco industry is correct or incorrect about the relative harmfulness between HTPs and cigarettes; but it is helpful to know if tobacco industry promotion and marketing does resonate with consumers. We have slightly revised the abstract to: “…if smokers who were exposed to HTP advertising hold beliefs that are consistent with marketing messages of lower harmfulness”.

  1. “Smokers who have been exposed to tobacco industry marketing about HTPs appear to hold beliefs that are consistent with tobacco industry advertising messages.” It is not possible to say from the data here if these smokers changed their beliefs because they were exposed to marketing or if e.g. they were better able to recall marketing because it was in line with their beliefs. As this is a cross-sectional survey, it is not possible to make such a directional statement. This also applies to the discussion and conclusion.

Authors’ Response: We have acknowledged this as a study limitation. We have added your suggested text as well (in red):

Page 16: “…First, because this study was cross-sectional, it cannot be used to demonstrate causality. For example, we cannot determine whether perceptions of harm preceded or followed product use. Similarly,  it is also not possible to determine if smokers changed their beliefs about the relative harm of HTPs compared to cigarettes because they were exposed to HTP marketing, or if they were better able to recall marketing because it was in line with their beliefs”.

We have also edited the abstract conclusion to be less presumptive about the effect of tobacco industry marketing: Abstract: “…. Smokers who have been exposed to HTP advertising were more likely to perceive HTPs as less harmful than cigarettes”.

Introduction

  1. ‘exploded onto the market’ strikes me as an unfortunate choice of words.

Authors’ Response: This has been changed to: HTPs were introduced...”

  1. It may be worth mentioning that e-cigarettes are not legal in Japan as this may be one factor in their success or in companies targeting the country specifically

Authors’ Response: The following has been added to the introduction on page 2:

“…The growing popularity of HTPs in Japan (accounting for 24.3% of Japan’s tobacco market in March 2019 [16]) may in part be due to the prohibition of nicotine e-cigarettes; thus, there is no competition between HTPs and e-cigarettes”.

Methods

  1. Please mention the minimum age for inclusion in the sample section (20 as suggested by the sociodemographics?)

Authors’ Response: We have included the minimum age for inclusion in the study on page 3 (Methods). It is indeed age 20+.

  1. For the HTP user groups, I would suggest just using the response options as labels (e.g. at least weekly versus less than weekly).

Authors’ Response: This has been edited in the methods, results, and Figure 2.

  1. Can you link the analyses more clearly to the aims please, i.e. which analysis was run to address each aim. Similarly, the results sections could be structured more clearly along the aims, e.g. using short versions of the aims as headings

Authors’ Response: We have edited the objectives so that they could be more clearly aligned with the statistical analyses and results section:

Page 3: “Thus, the objectives of this study were to examine: (1) smoker’s harm perceptions of HTPs relative to cigarettes; (2) differences in relative harm perceptions between: (i) exclusive smokers and smokers who use HTPs (concurrent users); (ii) concurrent users based on HTP use frequency (frequent vs. infrequent HTP users), and (iii) concurrent users based on HTP and smoking frequency; and (3) whether smokers’ beliefs about lower harmfulness are associated with exposure to HTP advertising”.

The objectives above have been added to the Analyses section (page 5), and headings have been used in the Results section that are in-line with the wording of the objectives.

Results

  1. In the supplemental figure, the arrows for exclusions each sit one box too low.

Authors’ Response: Supplemental Figure 1 (the study flow diagram) has been revised.

  1. Spotted a typo in table 1 for 'age group 30-39'

Authors’ Response: Thank-you for spotting this typo. It has been corrected in the table.

  1. In Figure 1, it would be interesting to see the full range of response options from much less harmful to more harmful

Authors’ Response: There are 5 response options for this variable. While we understand that it would be interesting to present all 5 levels, published papers have commonly used 3-4 levels (less harmful, equally harmful, more harmful (or equally/more harmful) and I don’t know). For example, see:

  • East et al: https://www.ncbi.nlm.nih.gov/pmc/articles/PMC6204576/
  • Majeed et al: https://www.ncbi.nlm.nih.gov/pmc/articles/PMC5373478/
  • Elton-Marshall et al: https://www.sciencedirect.com/science/article/pii/S0306460319305507?via%3Dihub
  • Gravely et al: https://pubmed.ncbi.nlm.nih.gov/32191332/?from_single_result=European+Adult+Smokers%27+Perceptions+of+the+Harmfulness+of+E-Cigarettes+Relative+to+Combustible+Cigarettes%3A+Cohort+Findings+From+the+2016+and+2018+EUREST-PLUS+ITC+Europe+Surveys

We have provided there data here for the Reviewer:

Frequency

Percent

Much less harmful than smoking cigarettes

397

11.0

Somewhat less harmful than smoking cigarettes

1326

36.7

Equally harmful to smoking cigarettes

867

24.0

Somewhat more harmful than smoking cigarettes

48

1.3

Much more harmful than smoking cigarettes

39

1.1

Refused

13

0.4

Don't know

923

25.6

Weighted regression analyses:

POH_REVIEW

Estimate

Standard Error

DF

t Value

Mean

Standard Error of Mean

Lower Mean

Upper Mean

Much less harmful than smoking cigarettes

-0.97

0.07

3592

-12.92

10.0%

0.6%

8.8%

11.2%

Somewhat less harmful than smoking cigarettes

0.35

0.05

3592

6.94

37.5%

1.0%

35.6%

39.4%

Equally harmful to smoking cigarettes

-0.06

0.06

3592

-1.08

24.8%

0.9%

23.1%

26.4%

Somewhat more harmful than smoking cigarettes

-3.57

0.17

3592

-20.6

0.7%

0.1%

0.5%

1.0%

Much more harmful than smoking cigarettes

-3.65

0.23

3592

-15.77

0.7%

0.2%

0.4%

1.0%

Don't know

0.97

0.07

3592

12.92

26.3%

0.9%

24.6%

28.0%

We would like to propose to keep the categories as they have been presented herein, particularly because of the low rate of “somewhat more harmful” and “much more harmful” responses.

Discussion

  1. Having pictures in the discussion of a scientific article is unusual and these do not seem to be crucial to the manuscript; I would strongly recommend removing Figures 4 and 5.

Authors’ Response: Figures 4 and 5 have been deleted from the manuscript.

  1. The text on marketing (p12 to 13) is quite long and detailed, I am not convinced all of it is relevant to the present manuscript.

Authors’ Response: We have cut out a fair amount of content in the Discussion about advertising in response to this request.

  1. What is the evidence that these products are not reduced risk products compared with cigarettes? The strong statements in the conclusion would need evidence to underpin them or need rewording.

Authors’ Response: While the harm reduction potential of certain alternative tobacco products (e.g., snus, e-cigarettes) has undergone scientific scrutiny, less is known about the absolute and relative health effects of HTPs, particularly the relative risk of concurrent cigarette-HTP use compared with exclusive smoking (Notably, the majority of HTP users are also smoking cigarettes Japan). Limited studies have shown HTP emissions contain higher concentrations of toxicants than observed for e-cigarettes, indicating that HTP use patterns warrant particular scrutiny. The authors also believe that the tobacco industry should not be allowed to market HTPs as reduced-risk products until they have permission to do so (e.g., PMI has not yet received approval for their IQOS MRTP application in the United States): See: Popova et al. Light and mild redux: heated tobacco products' reduced exposure claims are likely to be misunderstood as reduced risk claims. Tob Control 2018, 27, s87-s95.

                However, in light of the Reviewer’s comment, we have re-worded the Conclusion (page 16).
